# Atrial fibrillation in middle-aged athletes: Impact on left atrial, ventricular and exercise performance

Emily Vecchiarelli[1], Robert F. Bentley[1], Kim A. Connelly[2,3,4], Paul Dorian[2,3,4], Andrew Yan[2,3,4], Susanna Mak[5], Zion Sasson[5], Jack M. Goodman[1,5]*

1 Faculty of Kinesiology and Physical Education, University of Toronto, Toronto, Ontario, Canada, 2 Division of Cardiology, St. Michael's Hospital, Toronto, Ontario, Canada, 3 Keenan Research Centre of the Li Ka Shing Knowledge Institute, St. Michael's Hospital, Toronto, Ontario, Canada, 4 Department of Medicine, University of Toronto, Toronto, Ontario, Canada, 5 Division of Cardiology, Mt Sinai Hospital, Toronto, Ontario, Canada

* jack.goodman@utoronto.ca

**Data Availability Statement:** All relevant data are within the paper and its Supporting Information files.

**Funding:** This research was supported by a Canadian Institutes of Health Research Operating

## Abstract

High volume endurance training may increase the risk of paroxysmal atrial fibrillation (AF) in middle-aged athletes. Limited data are available describing the cardiovascular phenotype of middle-aged endurance athletes, or the impact of AF on atrial function and exercise performance performed in sinus rhythm. The purpose of this study was to characterize LA phasic function at rest and during exercise in athletes with paroxysmal AF, and to determine its impact on exercise performance. Fifteen endurance trained males (EA) (56 ± 5 years) without AF and 14 endurance trained males with paroxysmal AF (EA-AF) (55 ± 8 years) underwent echocardiography during cycle-ergometry at light and moderate intensities. Resting LA maximal volumes were similar between EA and EA-AF (30 ± 4 vs. 29 ± 8 ml/m², p = 0.50), and there were no differences in atrial electromechanical delay (AEMD). During moderate intensity exercise, EA-AF had reduced LA conduit (30 ± 6 vs. 40 ± 5 ml/m², p = 0.002) LA booster volumes (17 ± 5 vs. 21 ± 4 ml/m², p = 0.021), and reduced LV stroke volumes (100 ± 12 vs. 117 ± 16 ml, p = 0.007). These results demonstrate that exercise testing in athletes with AF unmasks evidence of adverse functional cardiac remodelling that may contribute to impaired exercise performance. It is unclear whether these functional alterations are the consequence of AF. Reductions in LA conduit volume, LA booster volume, and LV stroke volume during exercise may be helpful in clinical management and distinguishing pathologic from physiologic remodelling.

## Introduction

The cardio-protective effect of regular exercise [1] has been characterized as a linear, inverse dose-response relationship that exists between physical activity and mortality risk. However, the dose-response relationship may reflect a reverse 'J' [2], where optimal, recommended exercise reduces, but high volumes of exercise increases the risk of adverse outcomes, including the

Grant (JG, 130477) with additional financial support provided by the Heart and Stroke/Richard Lewar Centres of Excellence in Cardiovascular Research. EV received financial support from the Dr. Terry Kavanagh Fellowship, Faculty of Kinesiology and Physical Education, University of Toronto; KAC was supported by a Merit Award from the University of Toronto Temerity Faculty of Medicine and holds the Keenan chair in Research Leadership.

**Competing interests:** The authors have declared that no competing interests exist.

development of lone atrial fibrillation (AF) in middle-aged endurance athletes [3]. The European Society of Cardiology guidelines state that more than 1500 hours of lifetime vigorous exercise is a risk factor for AF in athletes, [4], and while some recent data suggests otherwise, [5] concerns of 'excessive' exercise and potential adverse cardiovascular outcomes in aging athletes persist. Moreover, there is a striking lack of evidence describing the cardiovascular phenotype in this population who develop lone AF and continue to exercise vigorously.

It is unknown whether aging athletes with paroxysmal AF, who have undergone successful radio frequency ablation and cessation of AF episodes, have diminished left atrial function or structure characteristics that may limit left ventricular (LV) output or degrade exercise performance. A greater understanding of how left atrial phasic function and structure impacts exercise performance in such athletes may improve our understanding of the pathophysiology of AF in athletes in addition to their long-term clinical management. The objective of this study was to characterize left atrial phasic function at rest and during submaximal exercise in endurance-trained middle-aged athletes with and without a history of paroxysmal AF. It was hypothesized that athletes with paroxysmal AF would have similar left atrial maximal volumes compared to healthy endurance athletes but compromised atrial reservoir function during exercise performed under conditions of sinus rhythm.

## Materials and methods

### Participants

Endurance-trained male athletes (45–65 years old) with vigorous year-round training were recruited for this study over a 6-month time period (January–June, 2018). Inclusion criteria required participation in endurance sport training for a minimum of 10 years in their respective activity (running, cycling, or multi-sport), history of regular competition (triathlon, marathons, long distance cycles of 100 km +), without a prolonged period of detraining. Exclusion criteria included history of cardiovascular disease (except paroxysmal AF), obstructive sleep apnea, thyroid disease, or diabetes. Athletes were recruited consecutively through advertisements and word of mouth at local clubs or referral by their treating physician over a period of 24 months. Healthy athlete recruits underwent medical screening including a full medical history, resting ECG and 48 hr Holter monitor. Presence of paroxysmal AF in athletes was verified previously by a clinical diagnosis by Holter or loop ECG. A $CHA_2DS_2VASC$ score was calculated for athletes with AF. In the absence of previous literature examining atrial phasic function in athletes with AF, a sample size calculation was completed based upon a cohort of middle-aged (~50 years old) individuals with lone AF compared to without [6]. Given an expected difference $6.5 \pm 5.2\%$ in left atrial reservoir strain with a power 0.8 and an alpha of 0.05, 11 individuals per group were required. This study was approved by the Research Ethics Board at the University of Toronto (#30988) in accordance with the Declaration of Helsinki, and all participants provided written informed consent prior to commencing study procedures.

### Patient and public involvement

Participants were not formally involved in providing input into study design, conduct, or scientific dissemination of results. A lay summary of the findings from this study (and relevant links) will be shared with participants and will be posted as part of an athlete and community education program on https://www.sportscardiologytoronto.com.

## Baseline participant characteristics and exercise training history

Baseline data including medical history, a 2-week exercise diary providing a summary of current training, and a modified Lifetime Physical Activity Questionnaire [7] to characterize long-term training history were collected. Athletes with AF completed additional screening to document their AF burden including typical frequency and episode duration. A training impulse (TRIMP) score [8] was calculated for each participant (weekly average of sessional exercise duration x rating of perceived exertion (RPE)). Resting measures of blood pressure (BP) (BpTRU model BPM-100, BpTRU Medical Devices, Coquitlam, Canada), height and body mass were obtained.

## Exercise protocol and echocardiography

Resting and submaximal exercise with concurrent echocardiography assessments was performed. After a recovery period of 30 minutes, a graded exercise test was completed. All participants refrained from caffeine for a minimum of 12 hours, food for 4 hours, and exercise 24 hours prior to testing. A standard 3-lead ECG was established for echocardiography assessments in which the skin was prepared with mild skin abrasion and isopropyl alcohol. Subjects were positioned semi-upright (35–45 degrees) on a tilt-recline stress echocardiography cycle ergometer (Ergoline 1200E, Ergoline, Germany) for optimal imaging quality. After a 5-minute rest period, resting echocardiography was performed, followed by two successive seven-minutes submaximal exercise stages designed to elicit heart rates (HRs) of approximately 100 (light stage) and 130 (moderate stage) bpm. Cardiac imaging, BP (Tango M2, Suntech Medical), and RPE were obtained during the last 2 minutes of each stage once at steady-state HR. Echocardiographic images were acquired by a single trained sonographer using a commercially available system (GE Vivid E9 Imaging System, GE Medical; Horten, Norway). Four chamber apical (A4C) images were obtained at a frame rate of 50–80 frames per second. Pulsed-wave Doppler interrogation of transmitral flow was obtained at the mitral leaflet tip level to determine early (E) and late (A) diastolic velocities. Tissue doppler recordings of mitral septal annular and lateral motion, and tricuspid annular motion were from the A4C. Early (E') and late (A') mitral lateral tissue doppler velocities were recorded. All images were analyzed offline by a single trained observer using dedicated software (GE Medical, EchoPac, Horten, Norway) and averaged over three cardiac cycles.

## Maximal exercise testing

Participants freely chose to complete a sport-specific (either cycle ergometer or treadmill) graded maximal exercise test to determine peak oxygen consumption ($VO_2$ peak) using a calibrated metabolic cart (Vmax Encore, CareFusion, YorbaLinda, CA). Treadmill testing followed a customized protocol described previously [9], as did cycle ergometry [10]. In both cases, $VO_2$ peak was determined at maximal effort, verified by a plateau in HR.

## Echocardiographic analysis

**Phasic volumes.** Left ventricular and atrial volumes were measured in A4C view [11]. Atrial volumes (Fig 1) were measured at different points of the cardiac cycle and phasic volumes (Fig 1) were determined as: reservoir volume (maximal volume–minimal volume), conduit volume (LV stroke volume–reservoir volume), passive emptying volume (maximal volume–pre-contractile volume), and booster volume as (pre-contractile volume–minimal volume). Atrial emptying fractions (EF) were calculated as follows: reservoir EF: [(maximal volume–minimal volume)/maximal volume] x 100, passive EF: [(maximal volume–pre

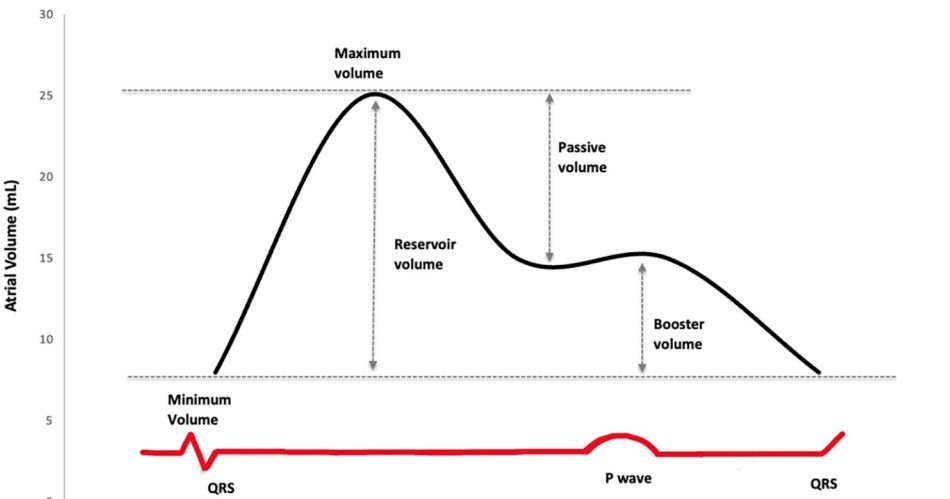

**Fig 1. Atrial time-volume curve and phasic volumes.** Conceptual illustration of the left atrial time-volume curve and phasic volumes as assessed in the present study.

contractile volume)/maximal volume] x 100, and booster EF: [(pre contractile volume–minimal volume)/minimal volume)/pre contractile volume] x 100. Absolute volumes were indexed to body surface area (BSA). LV stroke volume was computed as end diastolic–end systolic volume (EDV–ESV).

**Strain.** Left atrial and ventricular strain was assessed using Speckle-Tracking echocardiography, in which endocardial borders were traced using a point-and-click method on A4C images. Atrial phasic strain was calculated using the onset of the P wave as the reference point [12]. Strain curves were divided into phases: peak negative strain was identified as booster strain, peak positive strain as conduit strain, and the sum of the absolute negative and positive peak was identified as reservoir strain. Border tracking was previewed prior to processing to allow for visual confirmation of true endocardial tracking.

**Atrial electromechanical delay.** AEMD was measured at rest as previously described [13]. Pulsed Doppler A4C images with sample volume placed at the lateral and septal sides of the mitral annulus, and the right-ventricular tricuspid annulus were used. The time interval from P wave onset to the beginning of A' was measured from the lateral mitral annulus (lateral PA), septal mitral annulus (septal PA), and tricuspid annulus (tricuspid PA). Inter-AEMD was defined as lateral PA-tricuspid PA; intra-right as septal PA-tricuspid PA; and intra-left as lateral PA-septal PA.

All scans were analysed immediately post-collection by a single trained observer with an intra-observer intraclass correlation coefficient for atrial volumes, strain and AEMD (n = 10) of 0.98, 0.87 and 0.84, respectively.

## Statistical analysis

Statistical analyses were performed using SPSS Statistics software version 20 (IBM Inc.). Data was assessed for normality using a Shapiro-Wilk test. Demographic variables were assessed using independent samples t-tests and a Chi Square for ordinal data. A mixed model repeated measures analysis of variance (ANOVA) was used to analyze the a priori interaction of time (rest, light exercise, and moderate exercise) and group (EA and EA-AF). Significant F statistics with respect to the interaction term were followed by Bonferroni-corrected *post-hoc* t-tests. Resting AEMD was assessed with an un-paired t-test. Pearson correlations were performed

between atrial phasic volumes and LV function. A p-value of $<0.05$ was considered statistically significant. Data are presented as mean ± standard deviation (SD) along with ranges (min-max).

# Results

## Participant demographics and AF burden

Twenty-nine male endurance athletes completed all study protocols. All subjects with AF (n = 14) remained in sinus rhythm during exercise. Baseline characteristics for all participants are reported in Table 1. Healthy athletes were free of medication or history of premature atrial contractions or symptoms at rest or during exercise, and had no evidence of arrhythmias on resting ECG or 48 hour Holter monitoring. A total of four athletes with AF were identified to have a $CHA_2DS_2VASC$ greater than 0; three athletes had a cumulative score of 1 (due to hypertension), and one athlete had a cumulative score of 2 (due to hypertension and being 65 years of age). A total of 5 EA-AF participants were on a beta blocker, anticoagulant or calcium channel blocker, and one was also on an antiarrhythmic. In all cases, medications remained unchanged during all assessments. AF diagnosis was made on average, within 3 years prior to study recruitment. All AF athletes reported a variable time period of months to more than one year between symptom onset and securing assessment and diagnosis. Self-reported

**Table 1. Baseline participant characteristics.**

| Variable | EA | EA-AF | P value |
|---|---|---|---|
| | (n = 15) | (n = 14) | |
| Age (years) | 56 ± 5 (49–65) | 55± 8 (44–65) | 0.52 |
| Height (cm) | 180 ± 5 (171–188) | 177 ± 5 (170–189) | 0.18 |
| Weight (kg) | 76 ± 9 (62–91) | 80 ± 9 (64–95) | 0.20 |
| BSA ($m^2$) | 1.9 ± 0.1 (1.8–2.2) | 2.0 ± 0.1 (1.8–2.2) | 0.54 |
| BMI ($kg/m^2$) | 23 ± 2 (20–27) | 25 ± 2 (21–29) | **0.02** |
| SBP (mmHg) | 122 ± 17 (103–156) | 129 ± 12 (102–152) | 0.21 |
| DBP (mmHg) | 82 ± 9 (72–97) | 81 ± 9 (60–94) | 0.81 |
| HR (bpm) | 52 ± 8 (37–69) | 50 ± 7 (40–64) | 0.37 |
| $VO_2$ peak (L/min) | 3.9 ± 0.5 (3.3–4.7) | 3.8 ± 0.5 (3.1–4.5) | 0.46 |
| $VO_2$ peak (mL/kg/min) | 52 ± 5 (46–64) | 48 ± 6 (40–59) | 0.05 |
| 2 week average TRIMPs | 4020 ± 2151 (1554–9104) | 3021 ± 1750 (726–6731) | 0.19 |
| Total training hrs (hr/week) | 8.0 ± 4.3 (4–19.5) | 10.2 ± 6.2 (1.75–21) | 0.53 |
| Smoking history (n) | 5 | 2 | 0.39 |
| Alcoholic drinks (/week) | 7 ± 6 (0–21) | 5 ± 1 (0–14) | 0.30 |
| **Atrial Fibrillation Burden** | | | |
| Years since AF diagnosis | - | 4 ± 3 (1–10) | |
| Age at diagnosis | - | 51 ± 6 (43–60) | |
| History of atrial flutter (n) | - | 1 | |
| Past cardioversion (n) | - | 4 | |
| Past ablation (n) | - | 2 (both RA, within 1 to 4 years) | |
| AF episode triggered by Exercise | - | 8 | |

Data are presented as mean ± SD (range) unless otherwise noted. EA; endurance athletes, EA-AF; endurance athletes with atrial fibrillation, BSA; body surface area, BMI; body max index, DBP; diastolic blood pressure, HR; heart rate, SBP; systolic blood pressure, TRIMP; training impulse, $VO_2$ peak; peak rate of oxygen consumption.

paroxysmal episodes of AF occurred between once per month to daily episodes with a typical duration of minutes up to 12 hours before spontaneous resolution of sinus rhythm. Four AF patients had required cardioversion since diagnosis and two participants had undergone right sided atrial radio-frequency ablation. One athlete with atrial flutter was included. A similar absolute $VO_2$peak (L/min) was observed for the EA and EA-AF group, however when normalized for body mass, a smaller relative $VO_2$peak (mL/kg/min) was observed EA-AF group (p = 0.05).

### Atrial electromechanical function

There were no differences between EA and EA-AF in measures of inter-AEMD (33 ± 16 vs. 32 ± 7 ms, p = 0.81, intra-right (14 ± 8 vs. 12 ± 6 ms, p = 0.59), or intra-left (20 ± 10 vs. 20 ± 5 ms, p = 0.90).

### Resting volumes and cardiac function

HR, BP and LV resting data are summarized in Table 2. There were no significant differences in HR or BP at rest between groups. There were no differences in resting LV SV, EF, or GLS between groups. Similarly, there were no differences in LV diastolic function (E/A, E'/A', E/E') between groups.

Relative (indexed to BSA) atrial phasic function data is presented in Table 3 (for absolute volumes see S1 Table). Atrial phasic volumes and their relative (%) contribution to LV SV are presented in Fig 2. Resting maximal atrial volumes, atrial reservoir, passive and booster volumes were similar between EA and EA-AF. Resting conduit volume was greater in the EA (p = 0.04; Table 3). There were no differences between EA and EA-AF in resting atrial reservoir strain (30 ± 4 vs. 27 ± 6%, p = 0.31), passive strain (18 ± 4 vs. 16 ± 5%, p = 0.35), or booster strain (-12 ± 3 vs. -12 ± 3%, p = 0.66).

### Cardiac function during exercise

All participants were able to complete light and exercise protocols, reaching the target HR for both levels of intensity regardless of medication status in the EA-AF group, and there were no AF episodes the day of, or day before exercise assessments.

### Response to light exercise

There were no differences between EA and EA-AF in HR or BP at the onset of light exercise. There were also no significant differences in maximal atrial volumes, atrial reservoir, or passive volumes. There was an early increase in booster volume from rest to light exercise in EA (p = 0.003) which contributed to an increased in LV SV for EA. Conduit volume trended to be greater in EA compared to EA-AF (p = 0.058). Conduit volume was strongly correlated with LV SV (r = 0.803, p<0.001) for both EA and EA-AF. LV SV and GLS increased in both groups at onset of light exercise without significant difference between EA and EA-AF (Table 2). The contribution of conduit flow to LV SV remained constant and did not differ between groups (Fig 2). There were no differences in LV diastolic function between groups.

LA reservoir strain and booster strain were similar between EA and EA-AF during exercise (33 ± 8 vs. 35 ± 8%, p = 0.39; -27 ± 10 vs. -24 ± 10%, p = 0.45, respectively). Passive strain was reduced during light exercise in the EA (-6 ± 5 vs. -12 ± 8%, p = 0.04).

**Table 2. Left ventricular and blood pressure responses to exercise.**

| Variable | Rest | | | Light Exercise | | | Moderate Exercise | | |
|---|---|---|---|---|---|---|---|---|---|
| | EA | EA-AF | p | EA | EA-AF | p | EA | EA-AF | p |
| Workrate | | | | 112 ± 28 | 101 ± 25 | 0.31 | 173 ± 31 | 160 ± 28 | 0.28 |
| (Watts) | - | - | - | (70–160) | (70–150) | | (110–245) | (120–215) | |
| HR | 52 ± 8 | 50 ± 7 | 0.37 | 98 ± 5 | 99 ± 4 | 0.43 | 126 ± 6 | 123 ± 8 | 0.27 |
| (b/min) | (37–69) | (40–64) | | (90–104) | (93–105) | | (113–133) | (111–134) | |
| SBP | 122 ± 17 | 129 ± 12 | 0.21 | 174 ± 23 | 171 ± 31 | 0.90 | 206 ± 17 | 191 ± 31 | 0.30 |
| (mmHg) | (103–156) | (102–152) | | (123–215) | (132–218) | | (172–230) | (147–245) | |
| DBP | 82 ± 9 | 81 ± 9 | 0.81 | 79 ± 11 | 82 ± 18 | 0.65 | 84 ± 12 | 79 ± 21 | 0.63 |
| (mmHg) | (72–97) | (60–94) | | (62–95) | (49–118) | | (67–103) | (54–126) | |
| EDV | 78 ± 12 | 78 ± 14 | 0.930 | 82 ± 9 | 78 ± 11 | 0.334 | 82 ± 8 | 75 ± 12 | 0.151 |
| (mL/m$^2$) | (65–101) | (60–100) | | (70–101) | (63–94) | | (71–101) | (61–101) | |
| ESV | 34 ± 7 | 37 ± 10 | 0.473 | 28 ± 6 | 29 ± 7 | 0.757 | 23 ± 6 | 26 ± 9 | 0.454 |
| (mL/m$^2$) | (25–49) | (25–64) | | (19–42) | (18–44) | | (17–37) | (13–47) | |
| SV | 85 ± 15 | 79 ± 12 | 0.293 | 108 ± 16 | 97 ± 10 | 0.067 | 117 ± 16 | 100 ± 12 | **0.007** |
| (mL) | (64–112) | (65–103) | | (81–127) | (78–112) | | (85–137) | (75–123) | |
| EF | 55 ± 5 | 52 ± 7 | 0.195 | 67 ± 6 | 63 ± 5.0 | 0.173 | 72 ± 6 | 67 ± 8 | 0.120 |
| (%) | (48–67) | (36–62) | | (58–74) | (53–73) | | (62–79) | (54–80) | |
| GLS | -18 ± 3 | -18 ± 3 | 0.971 | -25 ± 4 | -23 ± 3 | 0.152 | -26 ± 3 | -23 ± 3 | 0.076 |
| (%) | (13–21) | (11–22) | | (18–33) | (17–27) | | (21–31) | (17–30) | |
| E/A | 1.3 ± 0.2 | 1.2 ± 0.2 | 0.240 | 1.5 ± 0.3 | 1.4 ± 0.3 | 0.550 | - | - | - |
| | (0.9–1.6) | (1.0–1.7) | | (0.8–1.9) | (1.0–1.9) | | | | |
| E'/A' | 1.5 ± 0.5 | 1.6 ± 0.5 | 0.710 | 1.5 ± 0.5 | 1.4 ± 0.4 | 0.330 | 1.2 ± 0.5 | 1.4 ± 0.5 | 0.250 |
| | (0.8–2.6) | (0.7–2.3) | | (1.0–2.5) | (0.9–2.4) | | (1.0–2.7) | (0.4–2.5) | |
| E/E' | 5.3 ± 1.1 | 4.8 ± 1.0 | 0.210 | 6.5 ± 1.4 | 6.3 ± 1.1 | 0.620 | - | - | - |
| | (3.8–7.7) | (4.1–7.5) | | (4.6–8.3) | (5.4–10.5) | | | | |

Data are means ± SD (range), with *p*-values denoting differences between groups at each workrate. SBP; systolic blood pressure, DBP; diastolic blood pressure, EA; endurance athletes, EA-AF; endurance athletes with atrial fibrillation, E/A; ratio of early to late diastolic filling velocities, E'/A'; ratio early to late mitral diastolic mitral annular velocity, E/E'; ratio of early filling to early diastolic mitral annular velocity, EDV; end diastolic volume, EF; ejection fraction, ESV; end systolic volume, GLS; left ventricular global longitudinal strain, HR; heart rate, SV; stroke volume.

### Response to moderate exercise

HR and BP remained similar between groups with the onset of moderate exercise. Atrial maximal, reservoir, and passive volumes did not differ between groups, though conduit (p = 0.002) and booster volumes (p = 0.021) were reduced in EA-AF compared to EA. LV SV was significantly lower in EA-AF (p = 0.007). The contribution of conduit flow to LV SV increased during moderate exercise for EA (p = 0.04) but not in EA-AF (Fig 2). Conduit volume remained strongly correlated with LV SV (r = 0.883, p<0.001). LV GLS and diastolic function remained similar between groups.

LA reservoir strain (37 ± 9 vs. 35 ± 8%, p = 0.60) passive strain (-0.3 ± 0.4 vs. -1.9 ± 3.9%, p = 0.24) and booster strain (-36 ± 9 vs. -33 ± 8%, p = 0.35) did not differ between EA and EA-AF.

## Discussion

To our knowledge, this is the first study to characterize exercise capacity, left atrial and left ventricular responses to exercise in athletes with a history of paroxysmal AF. Key findings are:

**Table 3. Relative left atrial volumes (indexed to BSA) and atrial phasic function.**

| Variable | Rest | | | Light | | | Moderate | | |
|---|---|---|---|---|---|---|---|---|---|
| | EA | EA-AF | *p* | EA | EA-AF | *p* | EA | EA-AF | *p* |
| Relative (mL/m²) | | | | | | | | | |
| LA$_{MAX}$ | 25 ± 5 | 26 ± 6 | *0.489* | 33 ± 5 | 30 ± 8 | *0.277* | 30 ± 4 | 29 ± 8 | *0.605* |
| | (16–31) | (19–36) | | (22–45) | (19–42) | | (23–36) | (18–41) | |
| LA$_{MIN}$ | 8 ± 3 | 9 ± 3 | *0.615* | 10 ± 5 | 10 ± 4.2 | *0.860* | 9 ± 3 | 10 ± 5 | *0.354* |
| | (5–13) | (6–15) | | (5–23) | (5–18) | | (4–15) | (2–17) | |
| LA$_{PRE-A}$ | 15 ± 4 | 15 ± 4 | *0.904* | 25 ± 5 | 19 ± 7 | ***0.024*** | 30 ± 3 | 27 ± 8 | *0.240* |
| | (9–20) | (10–23) | | (17–32) | (11–29) | | (23–36) | (14–40) | |
| LA$_{RES}$ | 17 ± 44 | 18 ± 3 | *0.547* | 23 ± 2 | 20 ± 5 | *0.088* | 22 ± 3 | 19 ± 4 | *0.085* |
| | (11–22) | (12–22) | | (15–25) | (14–28) | | (15–26) | (14–25) | |
| LA$_{PASS}$ | 10 ± 3 | 11 ± 3 | *0.398* | 8 ± 6 | 11 ± 6 | *0.228* | 1 ± 2 | 3 ± 3 | *0.075* |
| | (6–14) | (4–15) | | (0–16) | (3–20) | | (0–5) | (0–8) | |
| LA$_{COND}$ | 27 ± 6 | 21 ± 7 | *0.063* | 33 ± 5 | 28 ± 5 | *0.058* | 40 ± 5 | 30 ± 6 | ***0.002*** |
| | (16–35) | (12–33) | | (27–40) | (23–41) | | (30–48) | (20–44) | |
| LA$_{BOOST}$ | 7 ± 3 | 7 ± 2 | *0.720* | 15 ± 7 | 9 ± 4 | ***0.030*** | 21 ± 4 | 17 ± 5 | ***0.021*** |
| | (2–10) | (3–10) | | (5–24) | (4–18) | | (15–26) | (7–25) | |

Data are means ± SD (range), with *p*-values denoting differences between groups at each workrate. EA; endurance athletes, EA-AF; endurance athletes with atrial fibrillation, LA$_{MAX}$; Left atrial maximal volume, LA$_{MIN}$; left atrial minimal volume, LA$_{PRE-A}$; left atrial pre contractile volume, LA$_{RES}$; left atrial reservoir volume, LA$_{PASS}$; left atrial passive emptying volume, LA$_{COND}$; left atrial conduit volume, LA$_{BOOST}$; left atrial booster volume.

1) athletes with AF have similar LA reservoir volumes compared to healthy, age-matched endurance athletes, 2) atrial AEMD is similar between athletes with and without AF, and 3) athletes with paroxysmal AF have preserved atrial strain and LV diastolic function, but have smaller atrial booster, conduit and LV stroke volumes during submaximal exercise performed in sinus rhythm.

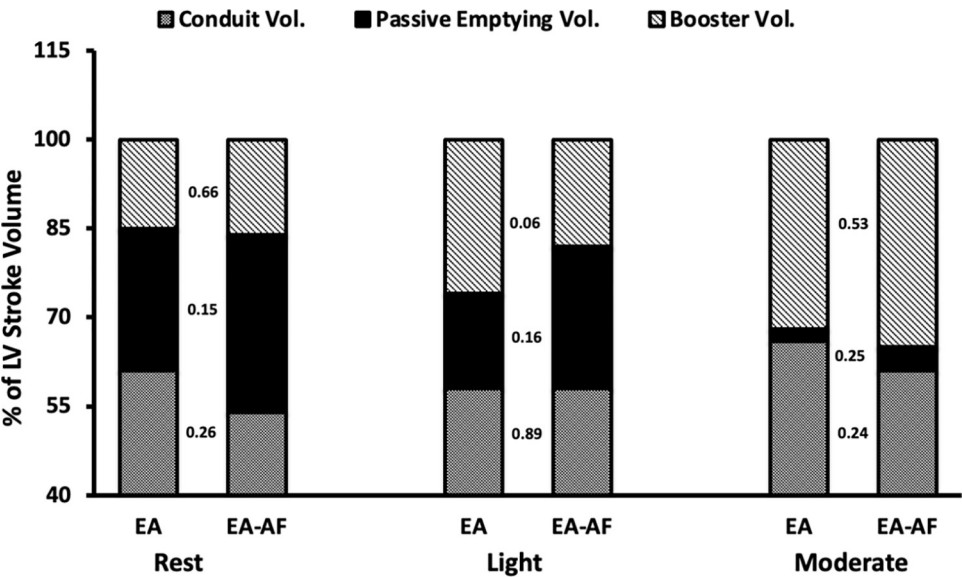

**Fig 2. Atrial phasic contribution to LV stroke volume.** Percent contribution of atrial phasic volumes to left ventricular (LV) stroke volume at rest, light, and moderate intensity exercise in athletes without (EA), and with atrial fibrillation (EA-AF). Values placed between bars denote group comparison *p*-value.

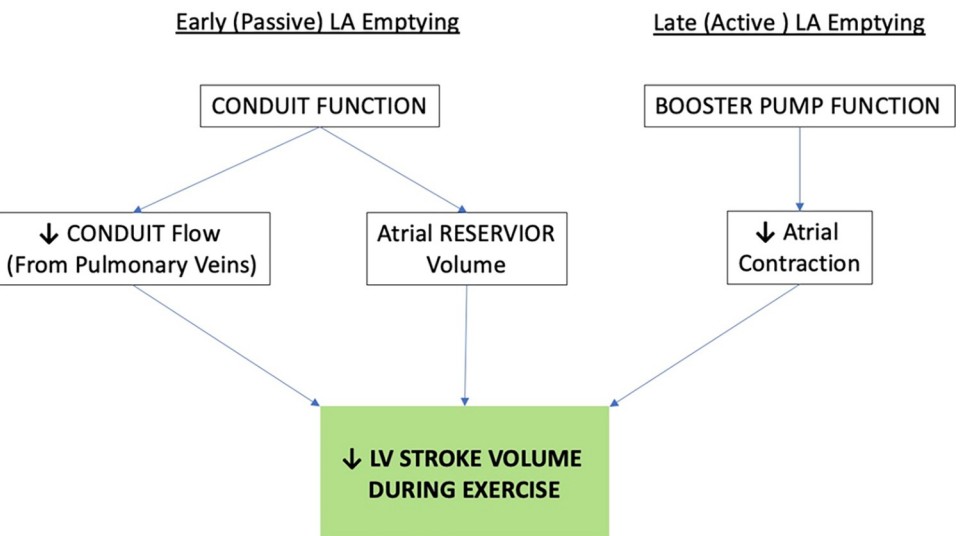

**Fig 3. Impact of atrial mechanics on exercise LV stroke volume.** Impact of left atrial (LA) mechanics on the stroke volume response to exercise. (Arrows indicate directional response in present study).

## Cardiac morphology

The present data supports the findings of Hubert et al. [14] which showed similar resting LV characteristics between age-matched veteran athletes with and without AF. We did not observe differences in resting LA maximal volumes between groups and our data is consistent with recent findings demonstrating similar LA maximal volumes in middle-aged athletes with and without paroxysmal AF [15]. All present athletes had indexed LA volumes within the normal range (dilation considered > 34 mL/m$^2$] [16], and given the similarity of lifetime training history, an overlap in atrial volumes was expected. A recent study exploring paroxysmal AF in older skiers in their seventies found larger atrial volumes in athletes with AF compared to those without [17]; however, larger atrial volumes in athletes without AF compared to non-athletes with AF have been reported [15] suggesting that atrial enlargement may not be the driving factor behind AF development in athletes. While future AF onset is possible in our healthy athletes, prospective longitudinal studies are required to further delineate the relationship between atrial size and AF risk in athletes.

We examined AEMD, a surrogate measure of atrial fibrosis linked to atrial impairment [18], on an exploratory basis since its prolongation may reflect disruption of electrical propagation across the atria independent of atrial size and may predict AF in patients with cardiovascular disease [13]. We did not observe differences in either AEMD or atrial size between EA and EA-AF, suggesting that the assessment of AEMD in athletes may have limited value. A large study is required to determine if AEMD offers has any predictive value in ascertaining risk for paroxysmal AF in athletes or has further prognostic insight once AF is diagnosed.

## Atrial contribution to LV function during exercise

A novel aspect of this study was the use of exercise to probe the cardiac response to sinus-rhythm exercise in athletes with paroxysmal AF. Left atrial function involves three distinct phases that modulate LV filling. During LV systole, the LA receives pulmonary venous return and acts as a reservoir, which is dependent upon LA relaxation to accommodate inflow at low pressures [19]. Early diastolic filling of the LV is facilitated by passive flow of LA reservoir

volume into the LV. Simultaneous to passive emptying, the rapid drop in ventricular pressure, generated partly by ventricular recoil, creates a suction effect that draws additional blood from the pulmonary veins into the LV. This phase is termed the 'conduit' component [20]. Upon late diastole, the LA actively contracts and acts as a 'booster' to eject remaining blood into the LV and complete EDV. Accordingly, passive emptying, conduit flow, and active emptying in combination determine LV SV, which was lower in the EA-AF group (Fig 2). Our group previously demonstrated the importance of these phases to ventricular filling during exercise in healthy middle-aged endurance athletes [21].

In non-athletic individuals with AF, LA reservoir volume may be reduced [22]. We failed to detect a reduction in LA reservoir function in EA-AF compared to EA, suggesting that LA compliance and relaxation was not compromised, congruent with recent data showing similar resting LA strain in athletes with or without AF [15]. However, we did observe reduced conduit and booster volumes at the onset of exercise in EA-AF. In healthy EA, enhanced Frank-Starling reserve (enhanced diastolic filling) during exercise allows for a continued, albeit slower rise in SV [23] despite increases in HR. This is largely dependent upon augmented conduit flow and LA booster function [21]. In athletes with AF, conduit volume plateaued during light exercise, which likely contributed to the lower SV (Fig 3). This is further supported by the correlation between conduit volumes and SV at both stages of submaximal exercise, which has previously been reported in inactive individuals [24] and individuals with paroxysmal AF [25]. In the present study, EA were able to recruit active atrial emptying at light exercise intensity to a greater extent compared to the EA-AF group, who in turn had more passive emptying, possibly due to higher diastolic left atrial pressures. It is possible that athletes are more dependent on passive emptying to maintain SV in the absence of booster function, particularly during exercise when episodes of AF occur [26]. In this scenario, exercise performance is likely to be reduced significantly. While atrial strain is considered a sensitive measure of atrial function [27], our inability to detect differences in strain between groups, despite reduced atrial conduit/booster volumes, may simply reflect a discordance between the extent of myocardial deformation and volumetric functional characteristics [28], or may simply be due to limited statistical power in detecting relatively small differences in atrial strain that are within normal limits.

The impact of increased LV stiffness on reduced passive LA emptying is less clear. Our data does not suggest that a reduced SV in EA-AF is due to a stiffer LV, as both LV GLS and standard measures of diastolic function (E/A, E'/A') were comparable between groups. The implications of the reduced LV SV in EA-AF is unclear. Since HR was targeted in our exercise protocol, it also remains unknown if the decline in SV would remain evident at higher HRs during exercise when LV filling time is greatly reduced. It is possible that at these intensity levels, a compromised SV would limit maximal cardiac output and therefore exercise performance in athletes with AF, even when performed in sinus rhythm, although we were slightly underpowered to detect this ($VO_2$ peak, $p = 0.05$). It is possible that measures of LV diastolic function (E/A, E'/A') may be insensitive to distinguish small differences in LV diastolic function between otherwise healthy athletes with and without AF, or if intrinsic reductions in LV diastolic relaxation contributes directly to diminished SV near peak exercise in athletes with AF. Anecdotal reports from most of our EA-AF participants suggest that regardless of their improved clinical state (e.g., successful ablation, medical treatment, etc.), development of AF attenuated training volume due to fear of recurrent AF episodes and this was corroborated by our observation of a lower training impulse (TRIMP). Further study is warranted to determine the long-term effects of AF-induced cardiac functional and structural remodelling.

## Clinical implications

The pathophysiology of AF in athletes remains poorly understood given the potential interaction of pathological and exercise-induced physiological remodelling [3]. Our cohort had a relatively low AF burden, including the frequency and duration of AF episodes, which usually spontaneously converts in athletes [29]. Notwithstanding, structural, functional and electrical remodelling of the atria and ventricles can be induced both by exercise training and AF independently, and these may have opposing effects. In sedentary populations, atrial enlargement is a predictor for AF [30] but may have limited prognostic value in identifying AF risk in athletes [31] given their propensity for physiological remodelling [32]. Additional risk factors for AF in non-athletic populations include prolonged P-wave duration [33], yet limited data in athletes with AF demonstrate normal P-wave morphology and duration [14], now corroborated by our data, suggesting it may lack sensitivity to identify athletes at risk for AF. However, the monitoring of atrial function may help with long-term management of athletes with AF. For example, diminished left atrial phasic function, which has been reported elsewhere in athletes [14] and non-athletes [34] with AF, may reflect progression of the AF substrate, worsening AF clinical status or increased risk of recurrence of AF episodes after ablation [35]. This progression may be of importance in athletes with AF who continue training, particularly at higher exercise intensities.

## Limitations

The paucity of data on atrial phasic function in healthy athletes or those with AF limited our ability to determine a sample size for our primary outcomes. As a result, our study may have been underpowered to detect subtle changes in left atrial reservoir strain between EA with and without AF. We also acknowledge that the sample size may limit generalizability, yet our study does provide novel insights of the functional response of the atria to exercise and impact on exercise performance. We note that several of the EA-AF participants reported a modest reduction in training intensity following diagnosis of AF which may have contributed to a higher BMI and lower relative VO2peak. While some degree of recent detraining may have occurred in the EA-AF group since AF diagnosis, lifetime exercise history and weekly exercise volumes were similar between groups. Given evidence that exercise-induced LV cavity remodelling can persist for years after exercise cessation [36], it is likely that any reduction in cardiac dimensions from attenuated training would be very small. We recognize the limitations of echocardiography and its propensity to underestimate cardiac volumes [11]. While this may have influenced the reported atrial absolute volumes, this imaging modality remains the most widely accessible and would not influence the interpretation of results in this experimental design. Additionally, while EA-AF athletes on negative inotropic medication achieved target heart rates during exercise, it is possible that atrial function was affected. We also recognize that upstream influences from the pulmonary arterial circuit may significantly influence left atrial function, but measuring pulmonary hemodynamics was not feasible. Lastly, we recognize the EA-AF was clinically heterogeneous in nature, both in terms of AF burden and interventions including atrial ablation. The impact of these factors on overall pathological atrial remodelling and function, remains unknown [37] and warrant further investigation.

## Conclusion

This study demonstrated that middle-aged endurance athletes with paroxysmal AF had similar LA maximal volumes and electro-mechanical function when compared to athletes matched for age and exercise history. During moderate exercise, atrial strain and LV diastolic function were preserved while atrial conduit and booster volumes were reduced in athletes with AF,

potentially compromising LV SV despite performing exercise whilst in sinus rhythm. The assessment of LV SV and LA conduit and booster volumes may be helpful in distinguishing physiologic from pathologic responses to exercise in athletes, particularly if these findings persist during higher exercise intensities and if clinical status includes a decrement in exercise performance. Further studies are warranted to confirm if continued training by athletes with AF impacts cardiac function and disease progression.

## Supporting information

**S1 Table. Absolute LA volumes and phasic function.** Data are mean ± SD (range). $LA_{MAX}$; Left atrial maximal volume, $LA_{MIN}$; left atrial minimal volume, $LA_{PRE-A}$; left atrial pre-contractile volume, $LA_{RES}$; left atrial reservoir volume, $LA_{PASS}$; left atrial passive emptying volume, $LA_{COND}$; left atrial conduit volume, $LA_{BOOST}$; left atrial booster volume. (DOCX)

## Acknowledgments

We would like to acknowledge Maggie Doherty and Meghan Glibbery who assisted in data collection for this project, and the research participants for their enthusiastic support.

## Author Contributions

**Conceptualization:** Jack M. Goodman.

**Data curation:** Emily Vecchiarelli, Robert F. Bentley.

**Formal analysis:** Emily Vecchiarelli, Robert F. Bentley.

**Funding acquisition:** Jack M. Goodman.

**Investigation:** Kim A. Connelly, Paul Dorian, Andrew Yan, Zion Sasson, Jack M. Goodman.

**Methodology:** Emily Vecchiarelli, Robert F. Bentley, Kim A. Connelly, Paul Dorian, Zion Sasson, Jack M. Goodman.

**Project administration:** Jack M. Goodman.

**Resources:** Kim A. Connelly, Paul Dorian, Jack M. Goodman.

**Supervision:** Robert F. Bentley, Jack M. Goodman.

**Writing – original draft:** Emily Vecchiarelli.

**Writing – review & editing:** Robert F. Bentley, Kim A. Connelly, Paul Dorian, Andrew Yan, Susanna Mak, Zion Sasson, Jack M. Goodman.

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
