## [Decision Letter · Decision Letter 0]

4 Dec 2023

PONE-D-23-31225Atrial Fibrillation in Middle-Aged Athletes: Impact on Left Atrial, Ventricular and Exercise PerformancePLOS ONE

Dear Dr. Goodman,

Thank you for submitting your manuscript to PLOS ONE. After careful consideration, we feel that it has merit but does not fully meet PLOS ONE’s publication criteria as it currently stands. Therefore, we invite you to submit a revised version of the manuscript that addresses the points raised during the review process.

We look forward to receiving your revised manuscript.

Kind regards,

Luigi Sciarra

Academic Editor

PLOS ONE

“This research was supported by a Canadian Institutes of Health Research Operating Grant (JG, 130477) with additional financial support provided by the Richard Lewar/Heart and Stroke Centres of Excellence in Cardiovascular Research. EV received financial support from the Dr. Terry Kavanagh Fellowship, Faculty of Kinesiology and Physical Education, University of Toronto; KAC was supported by a Merit Award from the University of Toronto Department of Medicine and holds the Keenan chair in Research Leadership.”

“This research was supported by a Canadian Institutes of Health Research Operating Grant (JG, 130477) with additional financial support provided by the Richard Lewar/Heart and Stroke Centres of Excellence in Cardiovascular Research. EV received financial support from the Dr. Terry Kavanagh Fellowship, Faculty of Kinesiology and Physical Education, University of Toronto; KAC was supported by a Merit Award from the University of Toronto Department of Medicine and holds the Keenan chair in Research Leadership.

We would like to acknowledge Maggie Doherty and Meghan Glibbery who assisted in data collection for this project, and the research participants for their enthusiastic support.”

“This research was supported by a Canadian Institutes of Health Research Operating Grant (JG, 130477) with additional financial support provided by the Richard Lewar/Heart and Stroke Centres of Excellence in Cardiovascular Research. EV received financial support from the Dr. Terry Kavanagh Fellowship, Faculty of Kinesiology and Physical Education, University of Toronto; KAC was supported by a Merit Award from the University of Toronto Department of Medicine and holds the Keenan chair in Research Leadership.”

Additional Editor Comments:

Dear authors, after a careful evaluation of your manuscript and according to reviewers' suggestions, our opinion is that it needs to be reviewed. It will be a pleasure to reconsider your paper after revision.

Best regards

Reviewers' comments:

Reviewer's Responses to Questions

**Comments to the Author**

1. Is the manuscript technically sound, and do the data support the conclusions?

Reviewer #1: Yes

Reviewer #2: Yes

2. Has the statistical analysis been performed appropriately and rigorously? 

Reviewer #1: Yes

Reviewer #2: Yes

3. Have the authors made all data underlying the findings in their manuscript fully available?

Reviewer #1: Yes

Reviewer #2: Yes

4. Is the manuscript presented in an intelligible fashion and written in standard English?

Reviewer #1: Yes

Reviewer #2: Yes

5. Review Comments to the Author

Reviewer #1: The manuscript by Vecchiarelli et al. is a research paper aiming to characterize left atrial phasic function at rest and during exercise in middle-aged endurance athletes with paroxysmal AF, and to determine its impact on exercise performance. The Authors conclude that exercise testing in athletes with AF unmasks evidence of adverse functional cardiac remodelling that may contribute to impaired exercise performance.

The topic is original and interesting. The paper is well written, the data are well collected, the results and conclusions are clear.

However, the paper shows some critical points:

Main points

-The presence of paroxysmal atrial fibrillation in athletes has previously been verified by clinical diagnosis using Holter or loop ECG. However, it is not described whether the presence of asymptomatic AF was tested in the control group, for example with ECG Holter. if it has not been done, it must be indicated in the methods and represents a limitation of the study.

-Four athletes with AF were treated with negative inotropic drugs and this may have influenced the value of ventricular stroke volume during exercise. It would be useful to evaluate the data only from athletes without therapy and discuss this aspect, also including it in the limitations of the study.

-Differences in VO2 peak between the two study groups (p 0,05) is not discussed in the text. Furthermore, it is not indicated how many patients performed a maximal test and therefore VO2 peak was assessed.

-Table 1: EA-AF have a higher BMI, this could be a relevant aspect and deserves a discussion.

-Table 3: LA pre-A is significantly lower in EA-AF athletes only in the mild exercise group. Is there a possible explanation for this data?

Minor points:

-Page 8: Inter-AEMD was defined as (lateral-tricuspid PA), intra-right as (septal PA-tricuspid PA), and intra-left as (lateral PA-septal PA). There's probably a typo, the brackets are not needed.

-Abstract: the AEMD acronym is missing.

Reviewer #2: Congratulations to the authors for the very interesting idea of the manuscript. The characterization of AF in athletes is an hot topic and deserves numerous studies to be explored.

However I have many concerns about this study:

The only difference in baseline characteristics between the two groups is represented by the BMI; for athletes it would be more appropriate to describe the percentage of muscle mass and fat mass, because we know that an elevated BMI, in non athlete population, is associated with an enhanced risk of developing AF, but an elevated BMI in athletes could be addressed to a relevant muscle mass and not to a overweight condition due to fat excess.

In addittion to a critical mass which is necessary as a substrate to initiate AF, atrial cardiomyopathy is required, and all the CHADVACS score components are a determinant of atrial cardiomyopathy; so you should add the CHADVASC Score for the group of partecipants;

Considering the characteristic of athlete population and their high rate of spontaneous AF cardioversion, you should include a reference about this topic, such as: “Mariani MV, Pierucci N, Piro A, Trivigno S, Chimenti C, Galardo G, Miraldi F, Vizza CD. Incidence and Determinants of Spontaneous Cardioversion of Early Onset Symptomatic Atrial Fibrillation. Medicina (Kaunas). 2022 Oct 24;58(11):1513. doi: 10.3390/medicina58111513.”

Moreover you should specify if any of them were on antiarrhythmic drugs.

All in all I believe this article fits for publication, after minor revisions.

6. PLOS authors have the option to publish the peer review history of their article (what does this mean?). If published, this will include your full peer review and any attached files.

Reviewer #1: No

Reviewer #2: No

---

## [Author Response · Author response to Decision Letter 0]

29 Dec 2023

We have adjusted the manuscript as requested, and any alterations to funding, etc.., has been noted in the letter to the editor.

We have attached a document (Response to Reviewers); below is a copy of that content.

Reviewer Comments:

Response to Reviewers

Reviewer #1: 

1. The presence of paroxysmal atrial fibrillation in athletes has previously been verified by clinical diagnosis using Holter or loop ECG. However, it is not described whether the presence of asymptomatic AF was tested in the control group, for example with ECG Holter. if it has not been done, it must be indicated in the methods and represents a limitation of the study.

We appreciate the reviewer’s comment. Healthy endurance athletes (EA) underwent medical screening for this reason and the methods section has been revised to include the following:

“Healthy athlete recruits underwent medical screening including a full medical history, resting ECG and 48 hr Holter monitor.”

While we cannot rule out AF with 100% certainty, the absence of any symptomatic history and ECG/Holter evidence of arrhythmias reflects a rigorous effort to ensure a very low likelihood of AF or cardiovascular disease in this cohort. We included this statement in the results:

“Healthy athletes were free of medication or history of premature atrial contractions or symptoms at rest or during exercise, and had no evidence of arrhythmias on resting ECG or 48 hour Holter monitoring.”

2. Four athletes with AF were treated with negative inotropic drugs and this may have influenced the value of ventricular stroke volume during exercise. It would be useful to evaluate the data only from athletes without therapy and discuss this aspect, also including it in the limitations of the study.

This is a valid concern. Our protocol used a target heart rate (HR) approach at both levels of exercise. We examined the individual response of patients on rate-control / negative inotropic medication (only 4 participants) and the presence of medication did not impede their ability to reach the HR targets. There is also some evidence that Beta blockers do not affect stroke volume at any given heart rate with training. (Mier CM, Domenick MA, Wilmore JH. Changes in stroke volume with beta-blockade before and after 10 days of exercise training in men and women. J Appl Physiol (1985). 1997 Nov;83(5):1660-5.)

We have also included data for mean HR achieved at each exercise stage for both groups of participants in Table 2, and there was no difference between groups. Notwithstanding we have inserted this statement into the exercise response section in the manuscript:

Results: 

Added Heading: “Cardiac Function During Exercise”

“All participants were able to complete light and exercise protocols, reaching the target HR for both levels of intensity regardless of medication status in the EA-AF group, and there were no AF episodes the day of, or day before exercise assessments”

Limitations:

“…while EA-AF athletes on negative inotropic medication achieved target heart rates during exercise, it is possible that atrial function was affected.”

3. Differences in VO2 peak between the two study groups (p 0,05) is not discussed in the text. Furthermore, it is not indicated how many patients performed a maximal test and therefore VO2 peak was assessed. 

Table 1: EA-AF have a higher BMI, this could be a relevant aspect and deserves a discussion.

These are valid concerns, and we agree that these findings warrant some elaboration in the manuscript. All participants completed all study protocols including maximal exercise testing. By convention, as with many studies that conduct maximal’ exercise testing, we report the peak value obtained, or “VO2peak”, given the low percentage of individuals who demonstrate a true maximal effort as defined by a plateau in oxygen consumption. As for BMI, all participants had BMIs in the normal range, well below levels associated with elevated AF risk. We did not measure fat or lean mass. We are not aware of specific links between these measures and AF risk, nor can we confirm with certainty if the lower relative VO2peak values (normalized for body mass) in the EA-AF group was due to a higher fat mass and therefore, higher BMI. It is possible that some degree of detraining since AF diagnosis contributed to this finding, yet data from long-term exercise histories remained very similar. These issues are now addressed directly in the Results and Discussion as outline below. 

Results: 

“Twenty-nine male endurance athletes completed all study protocols.”

Results:

“A similar absolute VO2peak (L/min) was observed for the EA and EA-AF group, however when normalized for body mass, a lower relative VO2peak (mL/kg/min) was observed EA-AF group (p=0.05).”

Limitations: 

“We note that several of the EA-AF participants reported a modest reduction in training intensity following diagnosis of AF which may have contributed to a higher BMI and lower relative VO2peak. While some degree of recent detraining may have occurred in the EA-AF group since AF diagnosis, lifetime exercise history and weekly exercise volumes were similar between groups. Given evidence that exercise-induced LV cavity remodelling can persist for years after exercise cessation (35), it is likely that any reduction in cardiac dimensions from attenuated training would be very small .”

4. Table 3: LA pre-A is significantly lower in EA-AF athletes only in the mild exercise group. Is there a possible explanation for this data? 

We appreciate the reviewer’s comment. The LA pre-A volume represents the residual volume in the atrium prior to atrial contraction and following passive flow of blood from the atrium to the left ventricle. Our athletes with AF had a smaller LA pre-A volume suggesting less volume remaining in the atrium for booster function/contractile output, which was reflected in both booster volumes at light and moderate exercise intensities. As included in the Discussion section, it is possible that our athletes with AF have more passive emptying (possibly due to higher diastolic left atrial pressures). These points are integrated into a revised passage within the Discussion:

“In the present study, EA were able to recruit active atrial emptying at light exercise intensity to a greater extent than the EA-AF group, who in turn had more passive emptying, possibly due to higher diastolic left atrial pressures. It is possible that athletes are more dependent on passive emptying to maintain SV in the absence of booster function, particularly during exercise when episodes of AF occur (26).”

Page 8: Inter-AEMD was defined as (lateral-tricuspid PA), intra-right as (septal PA-tricuspid PA), and intra-left as (lateral PA-septal PA). There's probably a typo, the brackets are not needed.

Thank you. This has been corrected.

5. Abstract: the AEMD acronym is missing. 

Thank you. This has been corrected.

Reviewer #2: 

1. Congratulations to the authors for the very interesting idea of the manuscript. The characterization of AF in athletes is an hot topic and deserves numerous studies to be explored.

However I have many concerns about this study:

The only difference in baseline characteristics between the two groups is represented by the BMI; for athletes it would be more appropriate to describe the percentage of muscle mass and fat mass, because we know that an elevated BMI, in non athlete population, is associated with an enhanced risk of developing AF, but an elevated BMI in athletes could be addressed to a relevant muscle mass and not to a overweight condition due to fat excess.

We have provided a response (and edited the manuscript) to address this concern raised by Reviewer 1, above. Despite the higher BMIs in the EA-AF group, the values are well within the normal range for this age-group and are not likely to have clinical relevance. It is possible the slightly higher BMIs are due to some degree of detraining but we did not measure fat or lean mass. The higher BMI likely contributed directly to the slightly lower “relative” VO2peak (normalized for body mass), but similar “absolute”, VO2peak (L/min). We thank the reviewer for these insights, and trust our changes (see above) address these concerns. 

2. In addition to a critical mass which is necessary as a substrate to initiate AF, atrial cardiomyopathy is required, and all the CHADVACS score components are a determinant of atrial cardiomyopathy; so you should add the CHADVASC Score for the group of participants;

Thank you for this important point; we have added the following to our Methods and Results sections: 

Methods: 

“A CHA2DS2VASC score was calculated for athletes with AF.”

Results:

“A total of four athletes with AF were identified to have a CHA2DS2VASC greater than 0; three athletes had a cumulative score of 1 (due to hypertension), and one athlete had a cumulative score of 2 (due to hypertension and age>65 yrs.).”

3. Considering the characteristic of athlete population and their high rate of spontaneous AF cardioversion, you should include a reference about this topic, such as: “Mariani MV, Pierucci N, Piro A, Trivigno S, Chimenti C, Galardo G, Miraldi F, Vizza CD. Incidence and Determinants of Spontaneous Cardioversion of Early Onset Symptomatic Atrial Fibrillation. Medicina (Kaunas). 2022 Oct 24;58(11):1513. doi: 10.3390/medicina58111513.”

Thank you for bringing this important paper to our attention. Our AF population was confined to those with paroxysmal AF who had a relatively low AF burden; none were in AF the day of or the day prior to the study. 

We have added the following to the discussion:

“Our cohort had a relatively low AF burden, including the frequency and duration of AF episodes, which usually spontaneously converts in athletes (Mariani et al).” 

4. Moreover you should specify if any of them were on antiarrhythmic drugs. 

This information is already present in the Results section, pg. 11 but has been simplified as follows:

“A total of 5 EA-AF participants were on a beta blocker, anticoagulant or calcium channel blocker, and one was also on an antiarrhythmic. In all cases, medications remained unchanged during all assessments.”

---

## [Decision Letter · Decision Letter 1]

15 Jan 2024

Atrial Fibrillation in Middle-Aged Athletes: Impact on Left Atrial, Ventricular and Exercise Performance

PONE-D-23-31225R1

Dear Dr. Goodman

We’re pleased to inform you that your manuscript has been judged scientifically suitable for publication and will be formally accepted for publication once it meets all outstanding technical requirements.

Kind regards,

Luigi Sciarra

Academic Editor

PLOS ONE

Additional Editor Comments (optional):

Reviewers' comments:

Reviewer's Responses to Questions

**Comments to the Author**

1. If the authors have adequately addressed your comments raised in a previous round of review and you feel that this manuscript is now acceptable for publication, you may indicate that here to bypass the “Comments to the Author” section, enter your conflict of interest statement in the “Confidential to Editor” section, and submit your "Accept" recommendation.

Reviewer #1: All comments have been addressed

Reviewer #2: All comments have been addressed

2. Is the manuscript technically sound, and do the data support the conclusions?

Reviewer #1: (No Response)

Reviewer #2: Yes

3. Has the statistical analysis been performed appropriately and rigorously? 

Reviewer #1: (No Response)

Reviewer #2: Yes

4. Have the authors made all data underlying the findings in their manuscript fully available?

Reviewer #1: (No Response)

Reviewer #2: Yes

5. Is the manuscript presented in an intelligible fashion and written in standard English?

Reviewer #1: (No Response)

Reviewer #2: Yes

6. Review Comments to the Author

Reviewer #1: (No Response)

Reviewer #2: The revised form of your paper is now suitable for publication; the methods are more clear now with the integration of the chadvasc value of the athletes.

7. PLOS authors have the option to publish the peer review history of their article (what does this mean?). If published, this will include your full peer review and any attached files.

Reviewer #1: No

Reviewer #2: No

---

## [Editor Report · Acceptance letter]

3 Mar 2024

PONE-D-23-31225R1 

PLOS ONE

Dear Dr. Goodman, 

I'm pleased to inform you that your manuscript has been deemed suitable for publication in PLOS ONE. Congratulations! Your manuscript is now being handed over to our production team.

Kind regards, 

on behalf of

Dr. Luigi Sciarra 

Academic Editor

PLOS ONE